# Explainable multi-task learning improves the parallel estimation of polygenic risk scores for many diseases through shared genetic basis

**Adrien Badré[1], Chongle Pan**[ID][1,2]*

**1** School of Computer Science, University of Oklahoma, Norman, Oklahoma, United States of America,
**2** Department of Microbiology and Plant Biology, University of Oklahoma, Norman, Oklahoma, United States of America

* cpan@ou.edu

## Abstract

Many complex diseases share common genetic determinants and are comorbid in a population. We hypothesized that the co-occurrences of diseases and their overlapping genetic etiology can be exploited to simultaneously improve multiple diseases' polygenic risk scores (PRS). This hypothesis was tested using a multi-task learning (MTL) approach based on an explainable neural network architecture. We found that parallel estimations of the PRS for 17 prevalent cancers in a pan-cancer MTL model were generally more accurate than independent estimations for individual cancers in comparable single-task learning (STL) models. Such performance improvement conferred by positive transfer learning was also observed consistently for 60 prevalent non-cancer diseases in a pan-disease MTL model. Interpretation of the MTL models revealed significant genetic correlations between the important sets of single nucleotide polymorphisms used by the neural network for PRS estimation. This suggested a well-connected network of diseases with shared genetic basis.

**Data Availability Statement:** All the code for data processing, model training, performance benchmarking, and model interpretation is

## Author summary

To prevent or delay the onset of complex diseases, an individual can benefit from knowing which diseases he/she is predisposed to through heredity and which diseases he/she is less susceptible to genetically. The overall genetic risk of a person to a complex disease is quantified using a polygenic risk score (PRS). Traditionally, PRS were developed independently for different diseases using statistical approaches. In this study, we used a multi-task learning approach and trained a deep learning model to simultaneously learn the PRS of many diseases all together. We showed that the new multi-task learning model can provide more accurate estimation of PRS for these diseases than their corresponding single-task learning models that were trained for individual diseases separately. The performance boost by multi-task learning suggests that many complex diseases may share a large number of common genetic risk variants among them, which can contribute to the positive transfer of knowledge during multi-task learning.

available publicly at https://github.com/thepanlab/GattacaNet2.

**Funding:** This project was supported by the startup fund to C.P. from the University of Oklahoma and an R01 grant (R01AT011618) to C.P. from the National Center for Complementary and Alternative Medicine and the National Institute of General Medical Science of the National Institutes of Health. The authors received financial support from the University of Oklahoma Libraries' Open Access Fund. The funders had no role in study design, data collection and analysis, decision to publish, or preparation of the manuscript. Both A.B. and C.P. received salaries from the University of Oklahoma.

**Competing interests:** The authors have declared that no competing interests exist.

## Introduction

The polygenic risk score (PRS) of a complex disease quantifies the genetic risk of an individual for this disease based on many genetic variants across the whole genome of this individual. The risk variants are generally selected based on this disease's genome-wide association studies (GWAS), often using a relaxed statistical significance threshold. A PRS can be estimated using a variety of statistical methods, including Best Linear Unbiased Prediction (BLUP) [1–5] and LDPred [6–9]. Statistical models of PRS have been built for breast cancer [8], colorectal cancer [10,11], Type-2 diabetes [12], cardiovascular disease [13], and many other diseases. These statistical methods generally assume that the effects of risk variants on a phenotype are linear and independent. Recently, machine learning approaches free of these assumptions [14] have been used to estimate the PRS for breast cancer [15], blood pressure [16], and schizophrenia [17]. However, the existing studies generally focused on constructing separate PRS models for individual diseases.

Many complex diseases share a substantial amount of common genetic determinants. Genome-wide cross-trait analyses have been performed between obesity and cardiovascular diseases [18], between thyroid and breast cancers [19], between uterine leiomyoma and breast cancer [20], between asthma and cardiovascular diseases [21], between Alzheimer's disease and gastrointestinal tract disorders [22], between Alzheimer disease and major depressive disorder [23], between lung cancer and chronic bronchitis [24], and so on. These studies were often motivated by frequent co-occurrences of pairs of diseases in a population. Some of the epidemiological associations have been attributed to the shared genetic architecture between the diseases. The related genetic etiology among diseases can be caused by dysfunctions in some common enzymes or pathways, which may increase the clinical risks for multiple diseases directly or indirectly.

In this study, we hypothesized that shared genetic determinants among diseases can be exploited to improve their PRS estimation. We tested this hypothesis using a pan-disease multi-task learning (MTL) approach [25] based on an interpretable neural network architecture [26]. MTL has been widely used in many computer vision [27] and natural text processing [28] applications, in which the training examples have multiple labels to be predicted from the same input feature vectors. Unlike single-task learning (STL) which trains a model to predict each individual label independently, MTL trains a model to predict all labels in parallel. MTL has been shown to provide better predictive performance than STL when the learning tasks are related [29]. Related tasks can enable a MTL model to learn a better shared representation through data amplification, feature selection, regularization, and other beneficial effects [30]. However, if the tasks are unrelated, the predictive performance of MTL may be worse than that of STL, owing to the negative knowledge transfer among the tasks [29]. Thus, if our hypothesis is invalid, the PRS learned for a disease in conjunction with other diseases by a pan-disease MTL model would be less accurate than the PRS learned for this disease by an STL model.

## Results

### Parallel prediction of many diseases by MTL

A neural network architecture was developed to predict many traits of an individual from their whole genome (Fig 1). This was an MTL extension of the linearizing neural network architecture (LINA) previously shown to provide good predictive performance for STL estimation of breast cancer PRS [26]. All the traits shared a latent genomic representation, which was an element-wise multiplication of a learned attention vector and the input whole-genome vector. Each trait was predicted from the shared representation via a task-specific hidden layer.

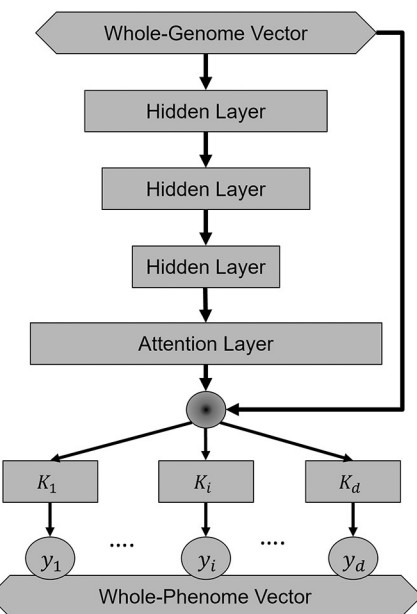

**Fig 1. An MTL deep neural network for parallel prediction of multiple traits.** This model was constructed based on the linearizing neural network architecture (LINA). The input layer (diamond box) contains the genotypes of genetic variants in a whole-genome vector. An attention vector is generated after 3 hidden layers (rectangular boxes) and then multiplied element-wise (round circle) with the input vector through a skip connection. The shared representation is used to predict each trait ($y_i$ in round circle). From end to end, a whole-phenome vector (diamond box) composed of many individual traits is predicted from this individual's whole-genome vector.

The output from the MTL model was a vector of character states for all the considered phenotypes, referred to as a whole-phenome vector.

Training a MTL model required a cohort of subjects with phenome-wide trait data. In this study, we used the United Kingdom Biobank (UKB) dataset and extracted 362 disease traits, including 69 cancer traits, from the electronic medical record of 488,175 UKB participants. Seventy seven diseases, including 17 types of cancers and 60 non-cancer diseases, had prevalence levels higher than 0.5% in the UKB cohort. We constructed two MTL models, one to predict the 69 cancers (pan-cancer MTL) and the other one to predict all 362 diseases (pan-disease MTL). Instead of selecting SNPs for each disease based on their statistical association, we included all 805,426 SNPs genotyped in the UKB cohort as the input for both MTL models. The UKB cohort was randomly divided into a training set (70%) for model training, a validation set (15%) for hyperparameter optimization, and a test set (15%) for performance benchmarking. A model's training took approximately 5 days on a computer node with dual A100 40GB GPUs. All the benchmarking results described below were based on the test set.

## Improved accuracy for PRS estimation by MTL

The estimation accuracy of malignant melanoma PRS was compared among STL, pan-cancer MTL, and pan-disease MTL (Fig 2). The same training data was used to train the STL model to predict malignant melanoma only (Fig 2A), the pan-cancer MTL model to predict 69 cancers, including malignant melanoma (Fig 2B), and the pan-disease MTL to predict malignant melanoma along with 361 other diseases (Fig 2C). The MTL and STL models generated different distributions of PRS in the test set. The differences were especially pronounced on the two shoulders of the distributions, which represented the subjects with higher or lower genetic

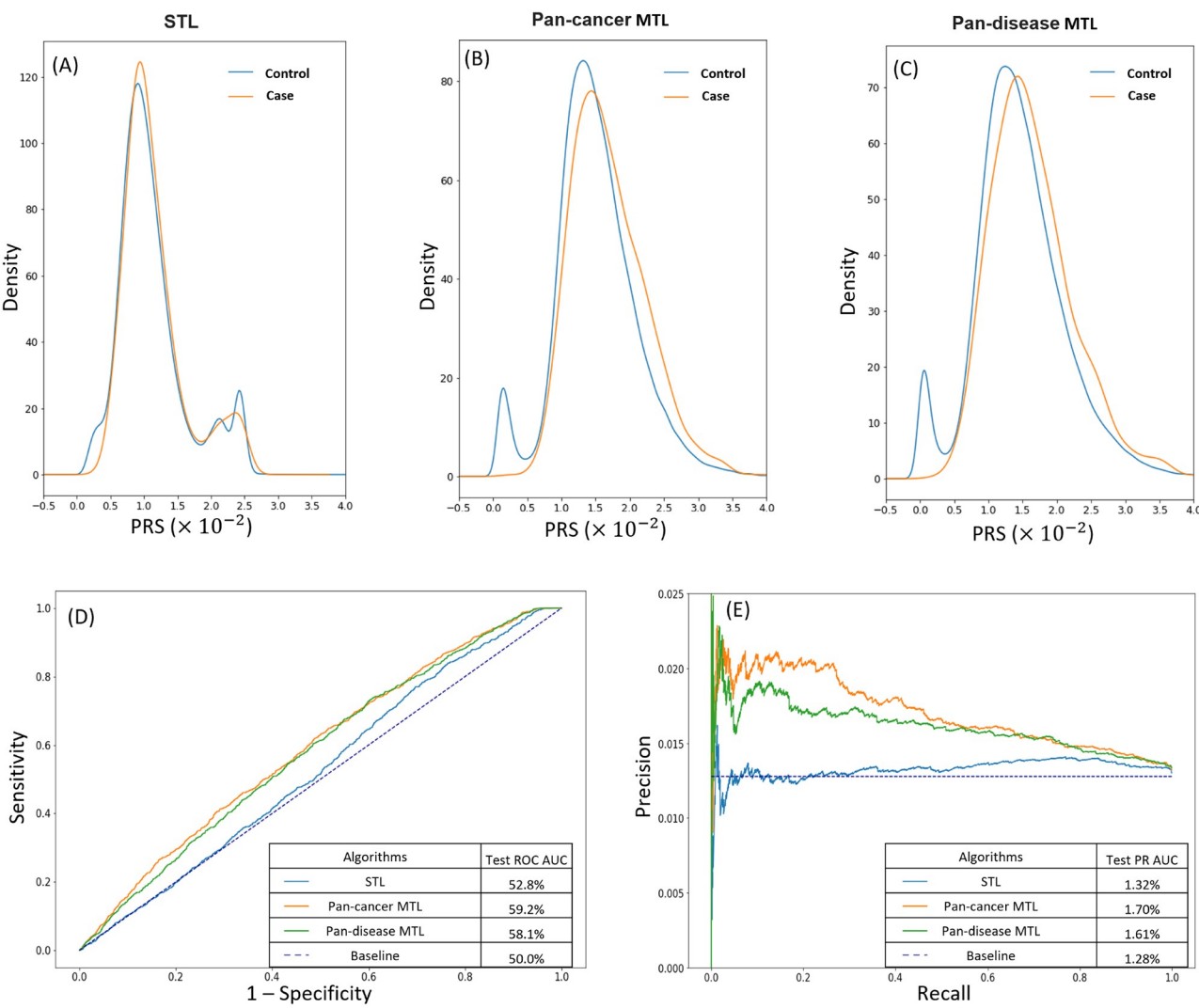

**Fig 2. PRS estimation for malignant melanoma by STL and MTL.** (A–C) Density plots of malignment melanoma PRS estimated by (A) STL, (B) pan-cancer MTL, and (C) pan-disease MTL. Each panel contains two overlapping density plots: a blue one for the control test cohort and an orange one for the case test cohort. The separation between the control and case density plots is greater in the two MTL panels than in the STL panel. (D) Receiver operating characteristic (ROC) curves of STL (blue), pan-cancer MTL (orange), and pan-disease MTL (green) for malignant melanoma PRS with the baseline (indigo dotted line). Both pan-cancer MTL and pan-disease MTL have larger ROC AUC than STL. (E) Precision-recall (PR) curves of STL (blue), pan-cancer MTL (orange), and pan-disease MTL (green) for malignant melanoma PRS with the disease prevalence as the baseline (indigo dotted line). The two MTL models also have larger PR AUC than STL.

risks than the average. The separation between the PRS distribution of the control subjects and the PRS distribution of the case subjects was greater in the two MTL models than the STL model. The predictive performances of the three models were compared using the Receiver Operating Characteristics (ROC) curves (Fig 2D). The Area Under the Curve (AUC) of the ROC curve by STL was only 2.8% higher than the 50% baseline, while those by the pan-cancer MTL and pan-disease MTL were 9.2% and 8.1% higher, respectively. Because of the imbalanced data, the predictive performances of the three models were also compared using the Precision-Recall (PR) curves (Fig 2E). Both MTL models achieved much higher precisions at the same recall level than the STL model. The baseline of the PR curve was determined by the

prevalence level of malignant melanoma in the UKB cohort, which was 1.28%. The PR AUC by STL was 0.04% higher than the baseline, while those by the pan-cancer MTL and pan-disease MTL were 0.42% and 0.33% higher, respectively. Overall, these metrics reflected better predictive performance of the two MTL models than the STL model for malignant melanoma.

The predictive performances of the two MTL models were then compared with the disease-specific STL models across 17 common cancers with prevalence levels higher than 0.5% (Table 1). The comparisons were made using both ROC AUC and PR AUC to account for the sensitivity, specificity, precision, and recall of the models. The two MTL models offered higher ROC AUC for 16 cancers and higher PR AUC for all 17 cancers than the disease-specific STL models. The magnitude of the performance improvement was quantified using the relative increase of the over-the-baseline AUC gain by an MTL model in comparison with the corresponding STL model. The average relative increase of ROC AUC over STL was +141% for the pan-cancer MTL and +153% for the pan-disease MTL. The average relative increase of PR AUC over STL was +96% for the pan-cancer MTL and +83% for the pan-disease MTL. The variability of the relative increases among different cancers suggested that each disease benefited to a different extent from MTL. The pan-cancer MTL had the highest ROC AUC for 4 cancers and highest PR AUC for 5 cancers. The pan-disease MTL had the highest ROC AUC for 12 cancer types and highest PR AUC for 12 cancer types. This suggested that the performance improvement from transfer learning increased with the number of traits in MTL.

**Table 1. Comparison of STL, pan-cancer MTL, and pan-disease MTL by ROC AUC and PR AUC for 17 cancer types with >0.5% prevalence.**

| Diseases | Receiver operating characteristics (ROC) AUC[#] | | | | | Precision-recall (PR) AUC[#] | | | | | Prevalence (Baseline) |
|---|---|---|---|---|---|---|---|---|---|---|---|
| | STL ROC AUC | Pan-cancer MTL | | Pan-disease MTL | | STL PR AUC | Pan-cancer MTL | | Pan-disease MTL | | |
| | | ROC AUC | Relative increase* | ROC AUC | Relative increase* | | PR AUC | Relative increase* | PR AUC | Relative increase* | |
| Malignant melanoma | 52.80% | **59.20%** | +234% | 58.10% | +194% | 1.33% | **1.70%** | +790% | 1.61% | +593% | 1.28% |
| Non-melanoma skin cancer | 61.50% | 62.40% | +8% | **62.90%** | +12% | 9.76% | 10.03% | +8% | **10.21%** | +14% | 6.65% |
| Skin cancer | 61.00% | 61.80% | +7% | **61.90%** | +8% | 10.40% | 10.73% | +11% | **10.86%** | +15% | 7.32% |
| Lung cancer | 59.10% | 60.30% | +14% | **60.50%** | +16% | 1.39% | 1.44% | +11% | **1.51%** | +23% | 0.90% |
| Intrathoracic cancer | 59.10% | 60.70% | +18% | **61.00%** | +21% | 1.54% | 1.58% | +7% | **1.65%** | +20% | 1.01% |
| Colorectal cancer | 54.40% | 56.40% | +46% | **57.10%** | +60% | 2.00% | 2.21% | +71% | **2.29%** | +100% | 1.72% |
| Colon cancer | 53.90% | 55.70% | +47% | **56.10%** | +59% | 1.38% | 1.49% | +51% | **1.49%** | +53% | 1.17% |
| Rectal cancer | 54.70% | 57.90% | +69% | **59.30%** | +100% | 0.77% | 0.88% | +98% | **0.89%** | +116% | 0.67% |
| Bladder cancer | 64.50% | 67.90% | +24% | **68.40%** | +27% | 0.80% | 0.87% | +24% | **0.92%** | +42% | 0.51% |
| Uterine cancer | 51.20% | **53.20%** | +177% | 51.80% | +50% | 1.08% | **1.18%** | +224% | 1.10% | +49% | 1.04% |
| Cervical cancer | 55.20% | 55.40% | +4% | **56.50%** | +24% | 1.80% | 1.88% | +35% | **1.97%** | +76% | 1.58% |
| Prostate cancer | **60.00%** | 59.70% | -3% | 59.60% | -4% | 8.33% | **8.53%** | +9% | 8.37% | +2% | 6.06% |
| Breast cancer | 57.00% | **58.30%** | +19% | 58.10% | +16% | 9.38% | 9.67% | +13% | **9.79%** | +20% | 7.25% |
| Female genital tract cancer | 54.00% | 54.30% | +7% | **54.50%** | +11% | 3.15% | 3.27% | +41% | **3.39%** | +84% | 2.86% |
| Male genital tract cancer | 53.60% | **56.10%** | +68% | 54.50% | +24% | 2.57% | **2.73%** | +57% | 2.59% | +9% | 2.28% |
| Lymphoma | 50.40% | 56.80% | +1442% | **57.90%** | +1704% | 0.73% | **0.82%** | +102% | 0.82% | +98% | 0.64% |
| Non-hodgkins lymphoma | 52.10% | 56.60% | +220% | **57.80%** | +278% | 0.61% | 0.68% | +81% | **0.69%** | +95% | 0.53% |

[#]Best AUC highlighted in bold

*Relative increase $= \frac{(MTL\ AUC - baseline\ AUC) - (STL\ AUC - baseline\ AUC)}{STL\ AUC - baseline\ AUC} \times 100\%$

To check whether the performance gain by MTL over STL can be generalized to non-cancer diseases, we compared the pan-disease MTL model with the disease-specific STL models for 60 non-cancer diseases with prevalence levels higher than 0.5% (S1 Table). The same set of performance metrics was used for the comparison. Compared with the disease-specific STL models, the pan-disease MTL model provided higher ROC AUC for 55 non-cancer diseases and higher PR AUC for 50 non-cancer diseases. The average relative increase by MTL across the 60 non-cancer diseases was +68% for ROC AUC and +82% for PR AUC. The benchmarking results for both cancer and non-cancer diseases indicated significant performance improvements by MTL over STL across many diseases.

## Identification of important SNPs for MTL by model interpretation

The first-order model-wise LINA interpretation algorithm [26] was used to identify the important SNPs used by MTL to predict each disease. A pan-cancer MTL model was trained and interpreted using an input whole-genome vector that contained the real SNPs and an equal number of decoy SNPs. Fig 3 shows the distributions of importance scores for the real SNPs and the decoy SNPs used by the MTL model to predict malignant melanoma. There were 59,350 real SNPs and 3,091 decoy SNP with important scores above $0.52 \times 10^{-3}$, which corresponded to a 5% FDR, because decoy SNPs with random association with the trait cannot be truly important for prediction. At the estimated FDR level of 0.1%, 48 real SNPs and no decoy SNP were identified as important for the MTL model to predict malignant melanoma. Many of these important SNPs have been identified as risk variants for melanoma in previous GWAS studies, including rs12203592 [31], rs62389423 [32], rs4785763 [33], rs4238833 [33], rs10931936 [34], rs1126809 [34], and Affx-35293625 [35].

Important SNPs in the pan-cancer MTL model were identified for the 17 prevalent cancers at the FDR levels of 0.1% and 5% (Table 2). The number of important SNPs at 0.1% FDR was 29 on average across the 17 cancers with substantial variability. These important SNPs may have strong associations with the traits. At 5% FDR, an average of 36,048 important SNPs were

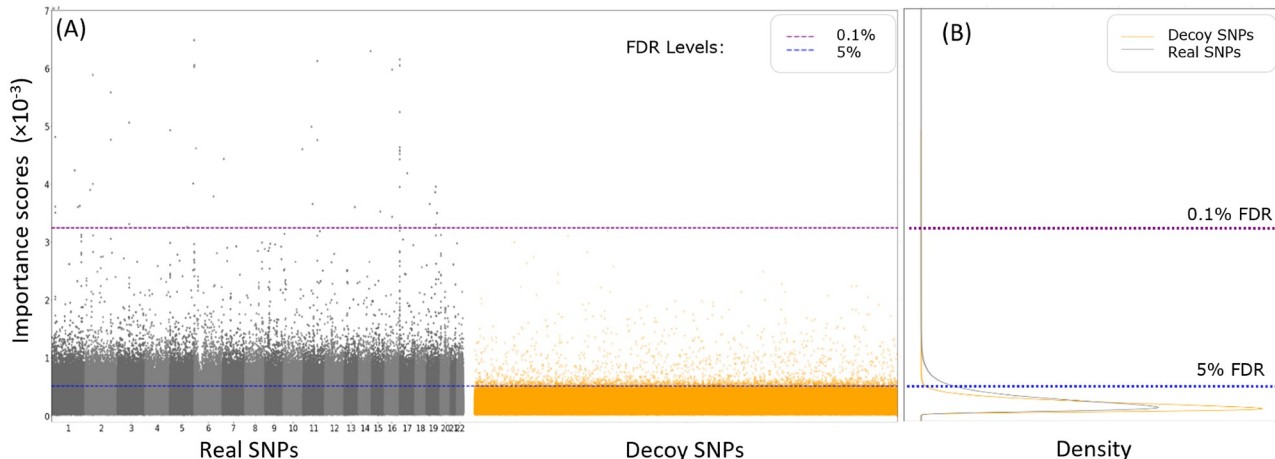

**Fig 3. Importance scores of real and decoy SNPs for malignant melanoma PRS estimation by pan-cancer MTL.** (A) Manhattan plots of real SNPs (black and grey dots) and decoy SNPs (orange dots) by their importance scores. (B) density plots of the importance scores of real SNPs (black curve) and decoy SNPs (orange curve). An estimated FDR of 5% (3091 decoy SNP to 59,350 real SNPs) was reached at the importance score threshold of $0.52 \times 10^{-3}$ (blue dotted line). No decoy SNPs and 48 real SNPs had importance scores above the threshold of $3.25 \times 10^{-3}$ for an estimated 0.1% FDR (purple dotted line).

**Table 2. Numbers of important SNPs used by pan-cancer MTL to estimate PRS of prevalent cancers.**

| Disease | FDR levels | 0.1% | 5.0% |
|---|---|---|---|
| Malignant melanoma | | 48 | 59350 |
| Non-melanoma skin cancer | | 132 | 48848 |
| Skin cancer | | 106 | 48419 |
| Lung cancer | | 4 | 41075 |
| Intrathoracic cancer | | 3 | 40392 |
| Colorectal cancer | | 36 | 45450 |
| Colon cancer | | 22 | 37487 |
| Rectal cancer | | 28 | 47904 |
| Bladder cancer | | 8 | 37 |
| Uterine cancer | | 25 | 38474 |
| Cervical cancer | | 5 | 42068 |
| Prostate cancer | | 23 | 94 |
| Breast cancer | | 34 | 96 |
| Female genital tract cancer | | 15 | 37083 |
| Male genital tract cancer | | 5 | 40742 |
| Lymphoma | | 0 | 43412 |
| Non-hodgkins lymphoma | | 4 | 41889 |

identified for the cancers, suggesting the use of diffused weak association signals across the whole genome by MTL for trait prediction.

We investigated the overlaps among the important SNPs for different diseases. At 0.1% FDR, only 4 common SNPs were shared among uterine cancer's 25 important SNPs, colorectal cancer's 36 important SNPs, and malignant melanoma's 48 important SNPs (Fig 4A). The number of important SNPs in the intersection for every pair of diseases at 0.1% FDR were listed in S2 Table. The relatively small intersections between different cancers indicated distinct SNP sets with large effect sizes for different diseases. At 5% FDR, there were 21,041 common SNPs shared among uterine cancer's 38,474 important SNPs, colorectal cancer's 45,450 important SNPs, and malignant melanoma's 59,350 important SNPs (Fig 4B). Genetic correlations were computed between every pair of cancers based on their importance scores for the SNPs important for one of the diseases or both at 5% FDR (Table 3). The genetic correlations were 0.88 between breast cancer and uterine cancer and 0.89 between lung cancer and lymphoma. Overall, 184 pairs of diseases had positive correlation coefficients between 0.5 and 1.0, 97 pairs had positive correlation coefficients between 0 and 0.5, and only 8 pairs had negative correlation coefficients. This suggested that MTL identified and may have exploited extensive genetic correlations between diseases to achieve a positive knowledge transfer among diseases for parallel PRS estimation.

## Discussion

Learning many tasks together in a neural network model does not automatically provide performance boost for all tasks [30,36]. Negative knowledge transfer can occur between unrelated tasks and, thereby, degrade the performance of a MTL model for these tasks [37]. We did not assume *a priori* which sets of diseases might be genetically related and could benefit from MTL. By aggregating many diseases together, we discovered positive knowledge transfer for most of the prevalent diseases studied here. The extent of positive knowledge transfer was quantified for each disease based on the gain of predictive performance by MTL relative to

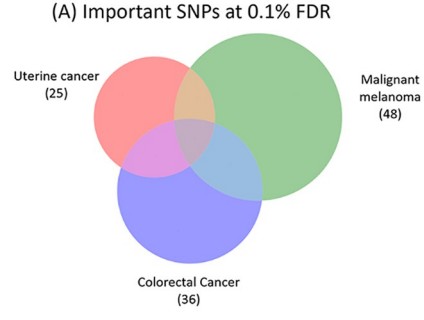

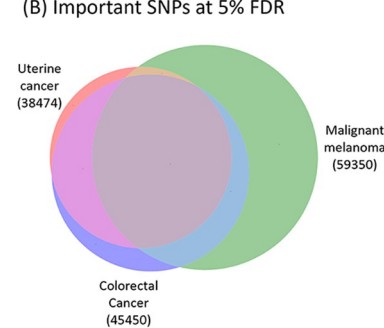

**Fig 4. Genetic correlation between diseases in MTL.** The Venn diagrams show the overlap among the important SNPs found for uterine cancer, malignant melanoma, and colorectal cancer at 0.1% FDR (A) and 5% FDR (B). The sizes of the circles and their overlaps are drawn proportionally. The important sets of SNPs for the three cancers have a small overlap at 0.1% FDR and a large overlap at 5% FDR.

STL. For example, malignment melanoma and uterine cancer benefited substantially from parallel training with the other cancers in the pan-cancer MTL, but the extent of positive knowledge transfer to the two cancers was reduced when many non-cancer diseases were added in the pan-disease MTL. The majority of prevalent cancers, including intrathoracic cancer, rectal cancer, and cervical cancer, gained additional performance by scaling MTL from 69 cancers to 362 diseases. Beneficial transfer learning was also evident for most of the non-cancer common diseases. Consistent observation of increased PRS accuracies for so many diseases provided strong support for the positive knowledge transfer during parallel learning of the genetic risks for complex diseases.

To understand how the PRS estimation benefited from MTL, we interpreted a pan-cancer MTL model and identified important SNPs for each cancer at two empirically estimated FDR levels. Many diseases shared a significant fraction of important SNPs at 5% FDR for their predictions. This suggested a beneficial joint selection of SNPs predictive of multiple diseases. This could be attributed to pleiotropy, wherein a genetic variant may have effects on multiple traits. A meta-analysis of many complex traits' GWAS results estimated 31% of the SNPs and 63% of the genes to be pleiotropic [38]. In addition, the joint feature selection in MTL may be better at filtering out SNPs with random trait associations in the training data than the disease-specific feature selection in STL can.

Data amplification may be a second mechanism for beneficial transfer learning in PRS estimation. Many diseases have an epidemiological correlation. For example, Woo et al.

**Table 3. Genetic correlation between important SNPs at 5% FDR.**

| Correlation Coefficient* | Non-melanoma skin cancer | Skin cancer | Lung cancer | Intrathoracic cancer | Colorectal cancer | Colon cancer | Rectal cancer | Bladder cancer | Uterine cancer | Cervical cancer | Prostate cancer | Breast cancer | Female genital tract cancer | Male genital tract cancer | Lymphoma | Non-hodgkins lymphoma |
|---|---|---|---|---|---|---|---|---|---|---|---|---|---|---|---|---|
| Malignant melanoma | 0.72 | 0.73 | 0.3 | 0.3 | 0.4 | 0.27 | 0.41 | 0.42 | 0.32 | 0.39 | 0.5 | 0.57 | 0.26 | 0.39 | 0.28 | 0.25 |
| Non-melanoma skin cancer | | 0.96 | 0.09 | 0.09 | 0.15 | 0.03 | 0.16 | 0.4 | 0.05 | 0.15 | 0.44 | 0.49 | 0.04 | 0.12 | 0.04 | 0.02 |
| Skin cancer | | | 0.05 | 0.05 | 0.12 | 0 | 0.13 | 0.38 | 0.03 | 0.11 | 0.42 | 0.48 | 0 | 0.09 | 0.01 | -0.01 |
| Lung cancer | | | | 0.99 | 0.86 | 0.86 | 0.86 | 0.74 | 0.78 | 0.84 | 0.91 | 0.87 | 0.86 | 0.8 | 0.89 | 0.89 |
| Intrathoracic cancer | | | | | 0.85 | 0.84 | 0.85 | 0.76 | 0.75 | 0.83 | 0.91 | 0.86 | 0.86 | 0.78 | 0.88 | 0.87 |
| Colorectal cancer | | | | | | 0.9 | 0.93 | 0.65 | 0.81 | 0.87 | 0.84 | 0.87 | 0.81 | 0.86 | 0.86 | 0.84 |
| Colon cancer | | | | | | | 0.83 | 0.69 | 0.83 | 0.81 | 0.87 | 0.87 | 0.85 | 0.81 | 0.9 | 0.9 |
| Rectal cancer | | | | | | | | 0.66 | 0.82 | 0.88 | 0.85 | 0.86 | 0.81 | 0.87 | 0.86 | 0.84 |
| Bladder cancer | | | | | | | | | 0.6 | 0.65 | -0.43 | -0.36 | 0.74 | 0.6 | 0.69 | 0.71 |
| Uterine cancer | | | | | | | | | | 0.8 | 0.77 | 0.88 | 0.78 | 0.82 | 0.85 | 0.84 |
| Cervical cancer | | | | | | | | | | | 0.82 | 0.89 | 0.86 | 0.81 | 0.84 | 0.82 |
| Prostate cancer | | | | | | | | | | | | -0.52 | 0.88 | 0.82 | 0.89 | 0.9 |
| Breast cancer | | | | | | | | | | | | | 0.88 | 0.82 | 0.89 | 0.88 |
| Female genital tract cancer | | | | | | | | | | | | | | 0.75 | 0.87 | 0.88 |
| Male genital tract cancer | | | | | | | | | | | | | | | 0.85 | 0.82 |
| Lymphoma | | | | | | | | | | | | | | | | 0.98 |
| Non-hodgkins lymphoma | | | | | | | | | | | | | | | | |

*The correlation coefficients are computed between the importance scores of the SNPs important for both or one of the two cancers at 5% FDR.

found a 75% greater risk of overall incident cancers after asthma diagnosis in adults [39]. Pooling the positive cases of multiple diseases together to train a MTL model may increase the effective sample size for learning a shared latent representation predictive of these diseases. Furthermore, many cancers may have some common genetic etiology. Pan-cancer risk variants may elevate the overall risk of individuals for cancers [40] and some environmental factors may determine the specific site of carcinogenesis. Pooling many cancer cases together may amplify the signal for discovering pan-cancer risk variants. Besides feature selection and data amplification, other mechanisms, such as eavesdropping, representation bias, and regularization [25], may also contribute to the positive knowledge transfer between diseases for PRS estimation.

Because hard parameter sharing was used in our neural networks from the input layer to the attention layer, the beneficial transfer learning may have produced a latent representation of the genomic data with better generalization for many diseases. Pervasive genetic correlations between diseases allowed MTL to improve the PRS estimation broadly across diseases. While many cross-strait studies have shown the genetic correlation between specific pairs of diseases [18–24], our study suggested that various degrees of shared genetic basis may be very prevalent among many complex diseases. Our results highlighted the potential value of holistic association studies between the whole human phenome and the whole human genome for both risk variant discovery and PRS estimation.

## Methods

### Preparation of the phenotypic and genomic data

A total of 488,175 subjects were extracted from the UK Biobank dataset release version 2 [41]. The phenotypic traits of the subjects were determined using the protocol and software described in a previous study [42]. The diseases in subjects were identified using hospital inpatient records (ICD10 codes, UK Biobank Data Coding 19) and self-reported disease status (UK Biobank Data Coding 3 for cancers and UK Biobank Data Coding 6 for non-cancer diseases). The UKB genomic data covered a total of 805,426 SNPs. The genotypes of SNPs were encoded as 0 for homozygous with the minor allele, 1 for heterozygous alleles, or 2 for homozygous with the dominant allele. All the code for data processing, model training, performance benchmarking, and model interpretation is available publicly at https://github.com/thepanlab/GattacaNet2.

### Construction of the MTL and STL models

The output of MTL LINA is a $d \times 1$ vector, $Y$, containing the predicted states of $d$ traits. The input of MTL LINA is an $m \times 1$ vector, $X$, containing the genotypes of $m$ SNPs. In this study, $d = 69$ in the pan-cancer MTL model, $d = 362$ in the pan-disease MTL model, and $m = 805,426$ in both models. MTL LINA can be expressed as:

$$Y = S(\boldsymbol{K} \cdot (A \circ X) + B),$$

$$A = F(X),$$

where $S()$ was a sigmoid activation function to be applied element-wise to its input column vector, $\boldsymbol{K}$ was a $d \times m$ coefficient matrix, $A$ was a $m \times 1$ attention vector, $B$ was a $d \times 1$ bias vector, $\cdot$ represented the matrix-vector multiplication, and $\circ$ represented the element-wise multiplication. $A$ was computed from $X$ by a feedforward neural network, $F()$, composed of 3 hidden layers containing 1000, 250, and 50 neurons. A leaky-ReLU activation function,

dropout with a dropout rate of 50%, and batch normalization were used in all three hidden layers. A linear activation function was used in the attention layer. $K$, $B$, and $F()$ were all learned from the training data.

The loss function of MTL LINA was defined as:

$$loss = W^T E + \beta ||K||_2$$

where $W$ was a $d \times 1$ vector of the loss weights for all traits, $E$ was a $d \times 1$ vector of the cross-entropy losses for all traits, and $||K||_2$ was the L2 norm of the coefficient matrix, and $\beta$ was the regularization weight. In this study, $W = [1, \ldots, 1]^T$ and $\beta = 10^{-3}$.

A total of 77 STL models were constructed for the 17 cancers and 60 non-cancer diseases with prevalence levels over 0.5%. All STL models used a feedforward neural network composed of three hidden layers containing 1000, 250, and 50 neurons as described previously [15]. A leaky-ReLU activation function, dropout with a dropout rate of 50%, and batch normalization were also used in all three hidden layers. The cross-entropy loss function was used to train the STL models.

## Training and benchmarking of the MTL and STL models

The 488,175 UKB subjects were randomly divided into a training set (70%), a validation set (15%), and a test set (15%). The training set was used to train all MTL and STL models by stochastic gradient descent. The training used mini-batches with a batch size of 512 and the Adam optimizer with an initial learning rate of $10^{-4}$. All MTL and STL models were trained for 100 epochs with checkpointing after every epoch. The checkpoints with the best performance on the validation set were kept for all MTL and STL models, which were the epoch-27 checkpoint for the pan-cancer MTL model and the epoch-25 checkpoint for the pan-disease MTL model. The training was carried out on a computer node with dual A100 40GB GPUs and 256 GB system memory. The training data was lazy-loaded to minimize memory usage using the pandas_plink [43] library. After the training was completed, the predictive performance of all MTL and STL models were benchmarked using the test set.

## Interpretation of the MTL models

The first-order model-wise interpretation was conducted to identify important features [26]. The Hessian of the output vector can be computed exactly from the Jacobian of the attention vector for the second-order interpretation as shown in the S1 Proof. Important feature interactions can be identified using the second-order model-wise interpretation.

A synthetic genomic vector was constructed for each subject to estimate the false discovery rate of the model interpretation, as shown previously [26]. The synthetic genomic vectors of all subjects contained all their real SNPs and an equal number of decoy SNPs. The genotypes of the decoy SNPs were randomly set to be 0, 1, or 2 with the same probabilities observed in the real SNPs. Thus, the decoy SNPs had identical frequencies of homozygous minor alleles, heterozygous alleles, and homozygous dominant alleles as the real SNPs. But, because the decoy SNPs should have no association with the phenotypes, any decoy SNP identified as important by the interpretation algorithm was considered a false positive hit.

A pan-cancer MTL model was constructed and trained as described above using the synthetic genomic vectors of the subjects in the training set. The importance scores of both real and decoy SNPs were computed for each cancer using the subjects in the test set. Only SNPs on the non-sex chromosomes were considered for model interpretation. The FDR for an importance score threshold was estimated as the ratio between the numbers of decoy SNPs to real SNPs above this threshold. The important SNPs at 0.1% FDR and 5% FDR were identified for all cancers with

>0.5% prevalence in the pan-cancer MTL model. The intersection and union of the important SNPs were counted between every pairs of prevalent cancers. The genetic correlation between two cancers was computed as the Spearman correlation coefficient between the importance scores of the SNPs belonging to the union of the SNP sets of the two cancers at 5% FDR.

## Supporting information

**S1 Table. Comparison of STL and pan-disease MTL by ROC AUC and PR AUC for 60 non-cancer diseases with >0.5% prevalence.**
(PDF)

**S2 Table. Numbers of shared important SNPs at 0.1% FDR between prevalent cancers in pan-cancer MTL.**
(PDF)

**S1 Proof. The Hessian of the output vector is the Jacobian of the attention vector for the second-order interpretation.**
(PDF)

## Acknowledgments

This research has been conducted using the UK Biobank Resource under Application Number 52970. We thank the high-performance computing resources provided by the University of Oklahoma Supercomputing Center for Education & Research.

## Author Contributions

**Conceptualization:** Adrien Badré, Chongle Pan.

**Data curation:** Adrien Badré.

**Formal analysis:** Adrien Badré.

**Funding acquisition:** Chongle Pan.

**Investigation:** Adrien Badré, Chongle Pan.

**Methodology:** Adrien Badré, Chongle Pan.

**Project administration:** Chongle Pan.

**Resources:** Adrien Badré, Chongle Pan.

**Software:** Adrien Badré, Chongle Pan.

**Supervision:** Chongle Pan.

**Validation:** Adrien Badré, Chongle Pan.

**Visualization:** Adrien Badré.

**Writing – original draft:** Adrien Badré, Chongle Pan.

**Writing – review & editing:** Adrien Badré, Chongle Pan.

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
