## [Decision Letter · Decision Letter 0]

23 May 2023

Dear Prof. Pan,

We are pleased to inform you that your manuscript 'Explainable multi-task learning improves the parallel estimation of polygenic risk scores for many diseases through shared genetic basis' has been provisionally accepted for publication in PLOS Computational Biology. We apologize for the delay in conveying this decision. As you will see, the paper was only formally reviewed by a single reviewer; however, members of the editorial board also reviewed the manuscript and concur with the accept decision.

Best regards,

William Noble

Section Editor

PLOS Computational Biology

Reviewer's Responses to Questions

**Comments to the Authors:**

Reviewer #1: This paper proposes an interesting hypothesis on exploiting the co-occurrences of diseases and their overlapping genetic etiology to enhance polygenic risk scores (PRS) for multiple diseases simultaneously using a multi-task learning (MTL) approach. The study utilized an explainable neural network architecture to build a pan-cancer MTL model and a pan-disease MTL model under their hypothesis. The authors observed improved accuracy in PRS estimation compared to individual single-task learning (STL) models in most of the cases. The outcomes of this paper suggest the potential of the co-occurrence and shared genetic determinants of diseases. Overall, this paper provides a compelling model and viewpoint for the potentials of MTL in improving disease risk prediction. The writings were smooth, and the experiments were explained well.

**Have the authors made all data and (if applicable) computational code underlying the findings in their manuscript fully available?**

Reviewer #1: None

PLOS authors have the option to publish the peer review history of their article (what does this mean?). If published, this will include your full peer review and any attached files.

Reviewer #1: No

---

## [Editor Report · Acceptance letter]

22 Jun 2023

PCOMPBIOL-D-23-00396 

Explainable multi-task learning improves the parallel estimation of polygenic risk scores for many diseases through shared genetic basis

Dear Dr Pan,

I am pleased to inform you that your manuscript has been formally accepted for publication in PLOS Computational Biology. Your manuscript is now with our production department and you will be notified of the publication date in due course.

With kind regards,

Zsuzsanna Gémesi
